# Effect of Vacuum Impregnation with Apple-Pear Juice on Content of Bioactive Compounds and Antioxidant Activity of Dried Chokeberry Fruit

**DOI:** 10.3390/foods9010108

**Published:** 2020-01-20

**Authors:** Agnieszka Nawirska-Olszańska, Marta Pasławska, Bogdan Stępień, Maciej Oziembłowski, Kinga Sala, Aleksandra Smorowska

**Affiliations:** 1Department of Fruit, Vegetable and Plant Nutraceutical Technology, Wroclaw University of Environmental and Life Sciences, Chełmońskiego Street 37, 51-630 Wrocław, Poland; agnieszka.nawirska-olszanska@upwr.edu.pl (A.N.-O.); aleksandra.smorowska@upwr.ed.pl (A.S.); 2Institute of Agricultural Engineering, Wroclaw University of Environmental and Life Sciences, Chełmońskiego Street 37a, 51-630 Wrocław, Poland; bogdan.stepien@upwr.edu.pl (B.S.); ksala195@gmail.com (K.S.); 3Department of Functional Food Products Development, Wroclaw University of Environmental and Life Sciences, Chełmońskiego Street 37, 51-630 Wrocław, Poland; maciej.oziemblowski@upwr.edu.pl

**Keywords:** *Aronia melanocarpa*, vacuum impregnation, polyphenolic compounds, sugars, ultraperformance liquid chromatography, photodiode array detection, tandem mass spectrometry

## Abstract

Food technology seeks ways to preserve products while maintaining high bioactive properties. Therefore, an attempt was made to assess the effect of the process of impregnation with apple-pear juice and the drying process on the content of bioactive compounds in chokeberry fruit. Chokeberry fruits were subjected to impregnation with apple-pear juice at three levels of vacuum pressure, 4, 6, and 8 kPa; then, they were dried using microwave-vacuum technology. The water activity of the obtained products, the content of fructose, glucose, sorbitol, and polyphenolic compounds, and antioxidant activity were determined. A total of 20 polyphenolic compounds were identified in the fruits and the obtained products (seven anthocyanins, six flavonols, four phenolic acids, and three flavan-3-ols). Preliminary processing, which consisted of introducing the juice ingredients into tissue of the chokeberry fruit, resulted in increased content of bioactive compounds. Moreover, a positive effect of impregnation on the antioxidant stability of the fruit after drying was noted. Water activity in the obtained products showed their microbiological safety. Impregnation at 4 kPa vacuum pressure proved to be the most desirable; in such conditions, the best product in terms of the content of bioactive compounds was obtained.

## 1. Introduction

Chokeberry belongs to the Rosaceae family and Pomoideae subfamily. This plant is native to North America, from where it was taken to Russia and then to other European countries, including Poland, where the largest plantations of this fruit can be found. Chokeberry cultivation does not require expenditure on crop-protection products since, due to their properties, these plants are immune to fungi and other pathogens. In a natural environment, chokeberry prefers humid forests and marshland [1].

The black chokeberry (*Aronia melanocarpa*, var. Nero) is a berry fruit that is rich in polyphenolic compounds, especially anthocyanins. Among plants, chokeberry fruits are characterised by one of the highest contents of anthocyanins and other polyphenols [2,3,4].

The chokeberry contains significantly more phenols, including anthocyanins, and shows stronger antioxidant activity than the blueberry, cranberry, and lingonberry [3]. The black chokeberry is a source of vitamins (P, B1, B2, B6, PP, and E), provitamin A (β-carotene), dietary fibre, minerals (Mo, Mn, B, J, Cu, Fe, Mg, and Ca), sugars (glucose, fructose, and sorbitol), and organic acids (citric acid, quinic acid, and malic acid) [5]. The most important group of compounds present in the chokeberry is that of polyphenols such as flavonoids (anthocyanins, flavonols, and flavan-3-ols) and phenolic acids (chlorogenic acid and neochlorogenic acid) [2,3,4,6]. The tart and bitter taste of chokeberry arises from the great number of polyphenols, mainly procyanidins, with a high degree of polymerisation. Procyanidin oligomers have a high affinity for proteins, causing their denaturation, which results in the feeling of tartness and dryness in the mouth as well as in choking [7].

Due to their tart taste, chokeberry fruits are not suitable for direct consumption. However, they are a valuable raw material used in the pharmaceutical and fruit and vegetable industries, as during processing, the tart–bitter taste becomes less intense. Chokeberry fruits are processed to obtain jam, juice, syrup, confitures, and jelly, and, due to the high content of anthocyanins, to also produce food colouring [5]. A nonperishable and convenient product that can be obtained from the whole chokeberry fruit is dried fruit, used as a snack or a tea additive and, in granulated form, as a diet supplement. The process of drying affects the chemical composition of the raw material, its appearance, and the quality and quantity of its bioactive compounds [8]. What is important in the process of drying is limiting the degradation of thermolabile compounds. Vacuum impregnation is an effective method of introducing any solution into the plant tissue. This process can be divided into two stages: initially, in conditions of lowered pressure, water and air are removed from the intercellular spaces of the material; then, the material is immersed in the impregnant and, under atmospheric pressure, the components of the liquid impregnant penetrate the tissue [9]. During vacuum impregnation, the free spaces are mechanically filled with the impregnant as a result of the pressure differential [10]. This method is applied in the preliminary processing of fruit and vegetables before drying. It is mainly used because of the possibility to introduce bioactive compounds and/or components enriching tastiness into the plant tissue [11].

A good method of drying fruit and vegetables containing nutrients that are valuable but nondurable at high temperatures, e.g., in the case of the chokeberry, is vacuum-microwave drying (VMD). This technique makes it possible to remove water in mild conditions with high intensity. Lowered pressure in the drying chamber decreases the boiling temperature of water, which enables the dehydration of the raw material at a temperature just slightly exceeding room temperature [12]. Due to the selection of appropriate microwave power, it is possible to control the temperature of the process, which protects plant tissue against damage [13]. In this method, heat is supplied to the raw material relatively fast. That increases the effectiveness of the inner transport of heat and shortens drying time as compared to other techniques [14]. Lowering the boiling temperature under conditions of lowered pressure reduces unfavourable changes in the structure, texture, and organoleptic features of the fruit, and prevents the loss of biologically active components [15].

The apple and pear fruits may not be too rich in bioactive compounds, but they are popular. Apples present a wide diversity of polyphenols that were classified into several major classes. Flavan-3-ols include monomeric (catechins) and polymeric (procyanidins) forms mainly constituted by (-)-epicatechin units. Among the hydroxycinnamic acids, 5-caffeoylquinic acid and 4-p-coumaroylquinic acid are present in the highest concentrations [16]. Apple juice is an important component of fruit intake in Europe. Most phytochemicals are not affected by applied juice-processing conditions, and juices can be considered as sources of putative bioactive compounds [17].

In the context of the above-mentioned facts, the aim of the present research was to assess the possibility of facilitating the production of dried chokeberry fruit using the process of impregnation with apple-pear juice. Such dried fruits contain a high content of bioactive compounds and are attractive to the consumer. In our study, the impregnated dried fruits were analysed in terms of their chemical composition (dry matter, ash, and sugars), content of polyphenols (by means of UPLC PDA-MS/MS) and antioxidant activity (by means of 2,2′-azino-bis-3-ethylbenzothiazoline-6-sulphonic acid (ABTS) and ferric reducing antioxidant potential (FRAP)). Additionally, water activity was measured as an indicator of microbiological safety. Apple-pear juice was used for impregnation to improve the tart taste of chokeberry fruits by introducing sugars, acids, aromas, and other sensoric components of apples and pears.

Studies on the health-promoting properties of chokeberry are in line with current trends and consumer interest in a healthy diet and lifestyle.

## 2. Material and Methods

### 2.1. Reagents and Standards

Acetonitrile, formic acid, methanol, ABTS, Trolox equivalents (6-hydroxy-2,5,7,8-tetramethylchroman-2-carboxylic acid), TPTZ (2,4,6-tris(2-pyridyl)-s-triazine), acetic acid, and phloroglucinol were purchased from Sigma-Aldrich (Steinheim, Germany); (+)-catechin, chlorogenic acid, neochlorogenic acid, cryptochlorogenic acid, 3-*O*-p-coumaroylquinic acid, cyanidin-*O*-galactoside, and cyanidin 3-*O*-glucoside were purchased from Extrasynthese (Lyon, France).

### 2.2. Plant Materials

Black chokeberry fruits were purchased from an organic farm (Wrocław, Poland) in 2018. Fruits of a standard size, and average diameter of 10 ± 2 mm, were cleaned, drained with tissue paper, and stored at 4 °C until processing; those with visible mechanical damage were excluded from the experiment.

### 2.3. Vacuum-Impregnation Treatment

Vacuum impregnation was carried out using laboratory installation VI-2016-MSP [11], located at Wroclaw University of Environmental and Life Sciences, Poland. As the infiltration liquid, freshly squeezed apple-pear juice (12.4° Bx), manufactured by Sady-Trzebnica Sp. z o.o. (Trzebnica, Poland), was used. A fresh fruit sample of 100 g was placed in the vacuum-impregnation chamber and held under vacuum pressure of 4, 6, or 8 kPa for 2 min at room temperature. The time of reaching the appropriate vacuum level was 30 s. Then, 600 mL of the impregnation juice was added (15 s), and infiltration under lowered pressure took place within 2 min. Next, the vacuum was released (15 s), and the sample was held in immersion under atmospheric pressure for 10 min. Total pretreatment time was 15 min. After that, the fruits were drained with tissue paper, weighed, and subjected to drying.

### 2.4. Vacuum-Microwave Drying

VMD under lowered pressure was carried out using laboratory installation SM 200 at the Institute of Agricultural Engineering, Wroclaw University of Environmental and Life Sciences. Pressure in the drying chamber fluctuated from 4 to 8 kPa. Six rotations of the chamber per minute were performed. The chokeberry fruits were drained in 60 g batches using a microwave power level of 240 W (4 W per 1 g of material). The drying process was repeated in triplicate.

### 2.5. Dry-Matter and Ash Content, and Water Activity

Assessment of the dry-matter content of the chokeberry fruit was performed by means of a gravimetric method [16]. The fresh chokeberry samples were precisely weighed (1.5 g) and dried at a temperature of 70 °C under 3 kPa vacuum pressure until a constant weight was obtained. Measurements were performed in triplicate and are expressed as %.

Fruit ash content was determined using the Association of Official Agriculture Chemists (AOAC) 930.09 method [18]. Measurements were performed in triplicate and are expressed as %.

Water activity was tested using a LabMaster-aw apparatus (Novasina AG, Lachen, Switzerland), with an accuracy of ±0.003, at a temperature of 25 ± 1.5 °C.

### 2.6. Antioxidant-Activity Analysis

The determination of the ABTS and FRAP content was performed in methanol extracts (80% *v/v*; material to extracting agent ratio was 1:5). ABTS and FRAP antioxidant assays were carried out as previously described by Benzie and Strain [19], and Re et al. [20], respectively, using a UV-2401 PC spectrophotometer (Shimadzu Corp., Kyoto, Japan). All assessments were performed in triplicate. Results are expressed in µmol Trolox equivalents/100 g dm.

### 2.7. Analysis of Sugars with HPLC-ELSD Method

The extract for sugar analysis was prepared as described by Oszmiański, Kolniak-Ostek, Lachowicz, Gorzelany, and Matłok [21]. Chromatographic analysis was carried out with an L-7455 liquid chromatograph (Merck-Hitachi, Tokyo, Japan) with an evaporative light scattering detector (PL-ELS 1000; Polymer Laboratories Ltd., Church Stretton, UK) and an L-7100 quaternary pump (Merck-Hitachi), equipped with a D-7000 HSM Multisolvent Delivery System (Merck-Hitachi), an L-7200 autosampler (Merck-Hitachi), and a Prevail Carbohydrate ES HPLC Column-W (250 × 4.6 mm, 5 µm; Alltech Inc., Nicholasville, KY, USA). Calibration curves (R2 = 0.9999) were created for glucose, fructose, and sorbitol. All data were obtained in triplicate. Results are expressed as grams per 100 g of dm.

### 2.8. Identification and Quantification of Polyphenols with UPLC-PDA-MS/MS Method

The extract for polyphenol analysis was prepared as described by Kolniak-Ostek and Oszmiański [22]. Polyphenol analysis was carried out using an ACQUITY UPLC system equipped with a binary solvent manager (Waters Corp., Milford, MA, USA), an ACQUITY UPLC BEH C18 Column (1.7 µm, 2.1 × 50 mm, Waters Corp., Milford, MA, USA), and a Q-Tof Micro mass spectrometer (Waters Corp., Manchester, UK) with an electrospray ionization (ESI) source operating in negative and positive modes. The mobile phase consisted of aqueous 0.1% formic acid (A) and 100% acetonitrile (B). Samples (10 µL) were eluted according to the linear gradient described previously by Oszmiański, Kolniak-Ostek and Biernat [23]. Mass spectrometry conditions were as follows: source block temperature, 130 °C; desolvation temperature, 350 °C; capillary voltage, 2.5 kV; cone voltage, 30 V; and desolvation gas (nitrogen) flow rate, 300 L/h.

Compounds were monitored at 280 nm (flavan-3-ols), 320 nm (phenolic acids), 340 nm (flavones), and 520 nm (anthocyanins). All experiments were done in triplicate. Results are expressed as milligrams per 100 g of dm.

### 2.9. Statistical Analysis

The obtained data were subjected to statistical analysis performed using Statistica v. 10.0 (StatSoft Polska, Kraków, Poland). They were recorded as means ± standard deviation (SD) and analysed by the Microsoft Excel 2007 software (Microsoft Corp., Redmond, WA, USA). Analysis of variance was performed with ANOVA procedures. Significant differences (*p* ≤ 0.05) between mean values were determined by Duncan’s multiple-range test.

## 3. Results and Discussion

Table 1 presents the dry-matter content, ash content, water activity, antioxidant activity, and the sugar content in chokeberry fruit: fresh, impregnated before drying, dried without previous impregnation, pretreated with impregnation under different vacuum pressure levels, and subjected to microwave drying.

The dry-matter content in the examined samples of dried fruit was comparable, and the samples constituted a statistically homogeneous group. Dry-matter-content values were similar to those obtained for chokeberry lyophilisates in the study by Oszmiański and Lachowicz [2].

Ash content was the greatest in fresh fruit not subjected to any processes. Impregnation and drying caused a decrease in ash content in the examined samples.

Water activity was determined in order to assess food susceptibility to microbial growth. The majority of micro-organisms develop when the water activity is above 0.95; the lowest water activity at which microbial growth can be observed is 0.6. Both in fresh fruit and in fruit impregnated without drying, water activity was very high (0.900 and 0.880, respectively), which is comparable to the results obtained by Samoticha, Wojdyło, and Lech [24]. Water activity determined for samples after drying was very low (0.144–0.207), although slightly higher in the case of nonimpregnated fruit (0.284). Water activity was slightly higher for negative pressure of 4 kPa (207), and the lowest for 6 kPa (144); this is a correct phenomenon. In the first phase of vacuum impregnation (operation of “dry” vacuum), air and native (free) water should be removed from intertissue spaces, while in the second phase, these spaces are filled with impregnation. This is what vacuum impregnation is about. The higher the vacuum level is, the better the effect of removing air and native water, and vice versa. Therefore, after impregnation at 4 kPa negative pressure, the highest water activity in the material was the most correct because more native water (active free water) was left in the tissue than at 6 and 8 kPa negative pressure. It follows that the impregnation process also contributed to the reduction of water activity in the tested samples. In samples impregnated and dried at different pressure levels, water activity was correlated with dry matter. Because water activity was less than 0.6, it protected fruits well against microbial growth. Impregnated dried fruit formed a statistically homogeneous group, whereas nonimpregnated dried fruit constituted a separate independent homogeneous group. The obtained results are comparable to those presented by Samoticha, Wojdyło, and Lech [24]. In their study, they used various drying methods (FD, freeze drying; VMD, vacuum-microwave drying; VD, vacuum drying; CD, convection drying; and CVM, convection-vacuum-microwave drying) and different drying temperatures and times, and the achieved water activity was in the range of 0.126–0.635.

Antioxidant activity was determined on the basis of cation radical ABTS* being quenched by antioxidants (ABTS assay) and the ability to reduce the Fe^3+^ ions (FRAP assay).

The greatest antioxidant activity in terms of ABTS* quenching was shown by fruit impregnated under vacuum pressure of 4 kPa (584.57 µmol Trolox equivalents/100 g dm); a lower value was obtained in the absence of impregnation (495.15 µmol Trolox equivalents/100 g dm). The results correspond to those reported by Teleszko and Wojdyło [25], with the antioxidant activity of fresh chokeberry fruit being 52.31 mmol Trolox equivalents/100 g dm. In previous research, slightly lower values were obtained by Oszmiański and Wojdyło [3]—439.49 µmol Trolox equivalents/100 g dm. Only impregnation and drying at pressure of 4 kPa contributed to the increase of antioxidant activity designated as ABTS. In other cases, antioxidant activity was lower than that measured in fresh fruit. The lowest one was in dried fruit without impregnation, which proves the positive effect of pretreatment as impregnation. The addition of apple-pear juice with its active compounds could have also contributed to high antioxidant activity.

The greatest antioxidant ability in terms of FRAP was observed in dried fruit impregnated under vacuum pressure of 4 kPa (289.47 µmol Trolox equivalents/100 g dm). During impregnation under vacuum pressure of 6 kPa, Fe^3+^ ion reduction activity was slightly lower (254.76 µmol Trolox equivalents/100 g dm), whereas the lowest FRAP value was observed for fruits without impregnation (123.12 µmol Trolox equivalents/100 g dm). It proves that preliminary processing with vacuum impregnation before drying had a beneficial effect on the antioxidant activity of the fruit, but only in conditions of 4 and 6 kPa vacuum pressure. At the lowest level of vacuum pressure (8 kPa), considerable changes in fruit structure and the degradation of bioactive compounds were observed.

Analysis of the sugar content in the studied material showed that the dominant sugar was sorbitol (from 39.56/100 g dm in nonimpregnated dried fruit to 43.98/100 g dm in fruit impregnated under vacuum pressure of 4 kPa). Glucose content was slightly higher than that of fructose (20.98–27.63/100 g dm and 19.18–22.63/100 g dm, respectively). In apple-pear juice, higher content of all sugars was determined and it was additionally determined that there were small amounts (0.83/100 g dm) of sucrose. The content of sugars apparently increased after impregnation with apple-pear juice as compared to fresh fruit, and then decreased after drying. This is due to the high content of sugars in apple-pear juice. The smallest amount of sugars was noted in nonimpregnated dried fruit, which means that, during impregnation, sugars from the juice were transported to the fruit. According to the conducted analyses, the process of drying resulted in a decreased content of sugars. In their research, Oszmiański and Lachowicz [2] obtained results significantly lower than the present ones; however, the tendency to a decreased content of sugars after drying is analogous.

Among the three indicated sugars, sorbitol was dominant in chokeberry. This is confirmed by other authors’ research [2,5].

### Comparison of Phenolic Compounds Detected in Chokeberry Products

Analysis results of the content of polyphenolic compounds in the examined samples are presented in Table 2. The particular compounds may be arranged according to their content in the following order: anthocyanins > phenolic acids > flavonols ≥ flavan-3-ols. An analogous scheme was obtained by Oszmiański and Lachowicz [2].

The main polyphenolic compounds in the studied products were anthocyanins (7 cyanidin derivatives), which constituted ~50% of polyphenols in general (Table 2). The dominant compound was cyanidin-3-galactoside. These compounds were not found in apple-pear juice. The total amount of anthocyanins in the examined samples ranged from 476.18 mg/100 g dm in nonimpregnated dried fruit to 1813.78 mg/100 g dm in fresh fruit. The impregnation significantly retained the anthocyanin content. The smallest loss of total anthocyanin content was noticed in drying at 4 kPa pressure and not much more in the test without drying. Despite the fact that apple-pear juice does not contain anthocyanins, its use significantly helped to preserve these compounds in dried chokeberry fruit. These values are significantly lower than those reported by Oszmiański and Lachowicz [2], who achieved from 6684.93 mg/100 g dm in dry powder of whole fruits (PDF) to 12,163.96 mg/100 g dm in dry powder from pomace crushed fruits (PPUF). In the present study, the total content of anthocyanins in the analysed samples ranged from 6.68/100 g dm in PDF to 12.16/100 g dm in PPUF.

Another studied group of compounds was phenolic acids (Table 2). On the basis of the UPLC technique, four phenolic acids were identified. The dominant compounds were chlorogenic acid and neochlorogenic acid; their content was 91.79–269.37 mg/100 g dm and 74.68–239.25 mg/100 g dm, respectively. The two other acids were present in trace amounts (0–11.21 mg/100 g dm). In the apple-pear juice 3-*O*-p-coumaroylquinic acid dominated, but was not a marked presence of neochlorogenic acid. It appears that 3-*O*-p-coumaroylquinic acid was impregnated with apple-pear juice since it was not present in fresh fruit before (Table 2). This was an additional positive effect of preimpregnation. Such a tendency is confirmed by other authors’ research [2,4]

Flavonols show antioxidant activity and play an important role in maintaining proper health [26]. In the studied chokeberry dried fruit, six flavonols were indicated. The content of quercetin-3-glucoside was the highest (9.61–30.86 mg/100 g dm), with the largest amount obtained in the case of impregnation under vacuum pressure of 6 kPa, and the smallest for nonimpregnated dried fruit. The process of impregnation without drying increased the content of flavonols, impregnation under vacuum pressure of 4 and 8 kPa with drying afterwards slightly decreased their content, whereas impregnation under vacuum pressure of 6 kPa resulted in insignificant rise in their content. A similar relationship was noted by Oszmiański and Lachowicz [2].

The last group of polyphenols present in chokeberry was that of flavan-3-ols. Three compounds were identified: quercetin-dihexoside, quercetin-3-*O*-vicianoside, and quercetin-3-*O*-robinobioside. Quercetin-3-*O*-vicianoside occurred in the largest amount (16.64–58.64 mg/100 g dm), while the content of quercetin-3-*O*-robinobioside was slightly lower (15.75–47.84 mg/100 g dm). Quercetin-3-*O*-vicianoside was the most abundant in fresh chokeberry fruit, and impregnation did not contribute to the growth of this flavan-3-ol, as was the case with other compounds from the flavan-3-ol group. The reason for this could be the complete lack of these compounds in apple-pear juice.

In apple-pear juice, procyanidins that had penetrated the impregnated chokeberry fruit were found in trace amounts or not at all. For all determined polyphenolic compounds, the largest loss compared to fresh fruit was observed during nonimpregnated drying. It seems that impregnation, as a preliminary process before drying, contributes to the protection of polyphenolic compounds. Anthocyanins and polyphenolic acids were protected in the best way during drying at 4 kPa. Flavanols and flavan-3-ols were protected in the most effective way at 6 kPa pressure.

## 4. Summary

Studies of dried chokeberry fruit saturated with apple-pear juice showed great protection of the bioactive properties of chokeberry fruit. 

In the studied dried chokeberry fruit, the dominant sugar was sorbitol; fructose and glucose were also detected, and the content of sugars depended on the method of impregnation. The low water activity of the obtained products was indicative of low microbiological activity; thus, the products could be regarded as safe.

The content of polyphenolic compounds in chokeberry is related to the studied variety, but it also resulted from the processes of impregnation and drying. The conducted research shows that impregnation with apple-pear juice contributed to an increase in polyphenol content except anthocyanins, which were not present in the impregnating juice, whereas the drying process had varied effects on their content.

Preliminary impregnation before drying proved to be beneficial, but only at the vacuum pressure levels of 4 and 6 kPa. Applying the lowest vacuum pressure of 8 kPa caused considerable changes in the structure of the fruit, as well as the degradation of bioactive compounds.

## Figures and Tables

**Table 1 foods-09-00108-t001:** Basic chemical composition of black chokeberry products.

	Fresh Chokeberry	Juice Apple-Pear	Impregnant, no Drying	Dried without Impregnation	4 kPa	6 kPa	8 kPa
Dry matter [%]	27.28 ± 1.02 ^b^	17.40 ± 1.09 ^c^	29.18 ± 1.41 ^b^	96.53 ± 1.74 ^a^	94.00 ± 1.67 ^a^	96.28 ± 1.11 ^a^	95.52 ± 1.41 ^a^
Ash [%]	2.93 ± 0.15 ^a^	2.11 ± 0.09 ^e^	2.57 ± 0.14 ^b^	2.48 ± 0.16 ^b^	2.36 ± 0.19 ^d^	2.42 ± 0.14 ^c^	2.35 ± 0.19 ^d^
aw [-]	584.57 ± 11.38 ^a^	-	0.880 ± 0.041 ^c^	0.284 ± 0.061 ^b^	0.207 ± 0.015 ^a^	0.144 ± 0.009 ^a^	0.172 ± 0.009 ^a^
ABTS [µmol/100 g dm]	567.82 ± 14.41 ^b^	519.15 ± 10.71 ^d^	533.94 ± 17.63 ^c^	495.15 ± 10.71 ^e^		539.71 ± 10.65 ^c^	521.89 ± 19.43 ^d^
FRAP [µmol/100 g dm]	218.36 ± 13.88 ^c^	189.19 ± 13.96 ^d^	204.06 ± 19.74 ^d^	123.12 ± 12.95 ^e^	289.47 ± 12.51 ^a^	254.76 ± 19.41 ^b^	190.82 ± 12.24 ^d^
Fructose	21.18 ± 1.18 ^d^	31.46 ± 2.32 ^a^	25.32 ± 1.95 ^b^	19.18 ± 1.03 ^e^	22.63 ± 1.33 ^c^	22.49 ± 1.46 ^c^	21.29 ± 1.72 ^d^
[g/100 g dm]							
Glucose	26.07 ± 1.19 ^c^	37.87 ± 3.14 ^a^	27.94 ± 1.58 ^b^	20.98 ± 1.36 ^d^	27.63 ± 1.22 ^b^	27.71 ± 1.54 ^b^	26.95 ± 1.63 ^b,c^
[g/100 g dm]							
Sorbitol	42.19 ± 2.03 ^c^	48.43 ± 3.56 ^a^	47.19 ± 2.17 ^a^	39.56 ± 1.99 ^d^	43.98 ± 2.37 ^b^	43.56 ± 2.72 ^b^	42.35 ± 2.32 ^b^
[g/100 g dm]							
Sucrose	0.00	0.83 ± 0.0 a	0.00	0.00	0.00	0.00	0.00
[g/100 g dm]							

ABTS, 2,2′-azino-bis-3-ethylbenzothiazoline-6-sulphonic acid; aw water activity; dm, dry matter; FRAP, ferric reducing antioxidant potential; mean values with different letters (a–e) within same row were statistically different (*p* = 0.05). Values expressed as mean ± standard deviation.

**Table 2 foods-09-00108-t002:** Comparison of phenolic compounds detected in black chokeberry products (mg/100 g dm).

Compounds ^1^	Rt [min]	λ Max [nm]	[H−M]^−^ (*m/z*) ^2^	Fresh Chokeberry	Juice Apple-Pear	Impregnant, no Drying	Dried without Impregnation	4 kPa	6 kPa	8 kPa
Anthocyanins										
Cyanidin-3.5-hexoside(epi)catechin	2.54	520	737+	16.95 ± 1.09 ^a^	0.00	15.59 ± 1.06 ^a^	5.96 ± 0.09 ^d^	15.71 ± 0.99 ^a^	14.20 ± 1.02 ^b^	10.04 ± 0.97 ^c^
Cyanidin-3-pentoside-(epi)catechin	2.98	520	707+	7.32 ± 0.48 ^a^	0.00	6.04 ± 0.34 ^b^	2.45 ± 0.19 ^d^	6.86 ± 0.28 ^a^	5.56 ± 0.46 ^b^	3.87 ± 0.28 ^c^
Cyanidin-3-hexoside-(epi)cat-(epi)cat	3.15	520	1025+	15.89 ± 1.41 ^a^	0.00	13.99 ± 1.20 ^b^	3.89 ± 0.27 ^e^	14.20 ± 1.23 ^b^	12.82 ± 1.08 ^c^	6.54 ± 0.54 ^d^
Cyanidin-3-galactoside	3.51	516	449+	1022.30 ± 99.4 ^a^	0.00	942.39 ± 91.2 ^b^	286.91 ± 13.6 ^d^	947.64 ± 87.3 ^b^	921.74 ± 82.9 ^b^	568.15 ± 46.3 ^c^
Cyanidin-3-glucoside	3.81	517	449+	49.76 ± 4.39 ^a^	0.00	45.64 ± 3.98 ^b^	11.93 ± 1.09 ^e^	46.79 ± 3.98 ^b^	41.15 ± 3.76 ^c^	24.31 ± 2.13 ^d^
Cyanidin-3-*O*-arabinoside	4.03	515	419+	611.64 ± 59.5 ^a^	0.00	515.28 ± 49.4 ^b^	142.57 ± 13.7 ^d^	521.14 ± 49.7 ^b^	498.06 ± 42.9 ^b^	264.34 ± 24.4 ^c^
Cyanidin-3-*O*-xyloside	4.68	515	419+	89.92 ± 7.94 ^a^	0.00	86.06 ± 7.89 ^b^	22.47 ± 1.93 ^e^	85.89 ± 7.53 ^b^	74.68 ± 6.93 ^c^	45.58 ± 4.12 ^d^
Sum	-	-	-	1813.78 ± 17.3 ^a^	0.00	1624.99 ± 15.7 ^b^	476.18 ± 3.54 ^e^	1638.23 ± 15.9 ^b^	1568.21 ± 13.9 ^c^	922.83 ± 8.98 ^d^
Phenolic acids										
Neochlorogenic acid	2.57	323	353	210.77 ± 20.9 ^b^	0.00	195.44 ± 18.9 ^b^	74.68 ± 6.98 ^c^	239.25 ± 22.9 ^a^	199.19 ± 18,6 ^b^	184.22 ± 17.3 ^c^
3-*O*-*p*-Coumaroylquinic acid	3.30	310	337	0.00	120.08 ± 11.3 ^a^	4.56 ± 0.34 ^b^	1.39 ± 0.09 ^c^	4.71 ± 0.37 ^b^	4.43 ± 039 ^b^	4.41 ± 0.41 ^b^
Chlorogenic acid	3.62	323	353	222.81 ± 21.2 ^b^	50.31 ± 4.21 ^d^	262.67 ± 24.7 ^a^	91.79 ± 9.1 ^c^	269.37 ± 24.9 ^a^	257.09 ± 23.7 ^a^	255.61 ± 24.7 ^a^
Cryptochlorogenic acid	3.71	323	353	4.83 ± 0.39 ^c^	1.89 ± 0.09 ^d^	7.68 ± 0.67 ^b^	2.54 ± 0.19 ^d^	11.21 ± 1.08 ^a^	11.03 ± 1.04 ^a^	10.74 ± 0.98 ^a^
Sum	-	-	-	438.41 ± 41.9 ^c^	64.28 ± 5.74 ^e^	470.35 ± 4.24 ^b^	170.40± 1.56 ^d^	524.54 ± 51.0 ^a^	471.74 ± 4.33 ^b^	454.98 ± 4.31 ^c^
Flavonols										
Quercetin-dihexoside	5.23	352	625	5.32 ± 0.46 ^c^	0.00	7.95 ± 0.67 ^a^	2.18 ± 0.17 ^d^	5.54 ± 0.46 ^c^	8.02 ± 0.79 ^a^	7.47 ± 0.67 ^a,b^
Quercetin-3-*O*-vicianoside	5.52	353	595	6.67 ± 0.61 ^c^	0.00	10.09 ± 0.99 ^a^	2.84 ± 0.07 ^d^	7.68 ± 0.65 ^c^	10.38 ± 0.98 ^a^	9.77 ± 0.65 ^a,b^
Quercetin-3-*O*-robinobioside	5.84	353	609	10.02 ± 0.99 ^c^	0.00	15.98 ± 1.36 ^a^	5.01 ± 0.29 ^d^	13.57 ± 1.21 ^b^	16.11 ± 1.34 ^a^	15.76 ± 1.47 ^a^
Quercetin-3-rutinoside	6.02	353	609	13.56 ± 1.29 ^c^	0.00	23.97 ± 2.07 ^a^	7.45 ± 0.69 ^d^	20.05 ± 19.1 ^b^	24.21 ± 2.37 ^a^	23.08 ± 2.24 ^a^
Quercetin-3-galactoside	6.09	352	463	17.47 ± 1.46 ^c^	0.64 ± 0.00 ^e^	25.64 ± 2.23 ^a^	8.31 ± 0.76 ^d^	20.54 ± 1.99 ^b^	25.97 ± 2.39 ^a^	25.47 ± 2.13 ^a^
Quercetin-3-glucoside	6.22	352	463	19.97 ± 1.53 ^c^	0.22 ± 0.00 ^e^	30.12 ± 2.97 ^a^	9.61 ± 3.98 ^d^	25.32 ± 2.38 ^b^	30.66 ± 2.98 ^a^	29.90 ± 2.56 ^a^
Sum	-	-	-	73.01 ± 6.45 ^b^	0.86 ± 0.00 ^d^	113.75 ± 10.9 ^a^	35.40 ± 2.89 ^c^	92.70 ± 8.97 ^a,b^	115.35 ± 10.3 ^a^	111.45 ± 10.5 ^a^
Flavan-3-ols										
Quercetin-dihexoside	5.29	352	625	16.31 ± 1.07 ^c^	0.00	21.24 ± 2.03 ^a^	8.21 ± 0.67 ^e^	14.79 ± 1.34 ^d^	21.73 ± 2.02 ^a^	19.19 ± 1.56 ^b^
Quercetin-3-*O*-vicianoside	5.50	353	595	62.09 ± 5.79 ^a^	0.00	58.44 ± 4.56 ^b^	16.64 ± 1.43 ^d^	42.77 ± 4.09 ^c^	58.64 ± 4.98 ^b^	57.01 ± 4.64 ^b^
Quercetin-3-*O*-robinobioside	5.87	353	609	30.24 ± 2.98 ^c^	0.00	47.62 ± 3.87 ^a^	15.75 ± 1.42 ^d^	39.96 ± 3.24 ^b^	47.84 ± 4.08 ^a^	46.61 ± 4.01 ^a^
Sum	-	-	-	108.64 ± 9,78 ^b^	0.00	127.30 ± 11.9 ^a^	40.60 ± 3.78 ^d^	97.52 ± 8.93 ^b,c^	128.21 ± 12.3 ^a^	122.81 ± 11.7 ^a^
Flavonols and Procyanidins										
Procyanidin B1	2.47	275	577	0.00	9.62 ± 3.0 ^a^	0.00	0.00	0.00	0.00	0.00
(+)-catechin	2.81	280	289	0.00	10.95 ± 1.1 ^a^	0.00	0.00	0.00	0.00	0.00
Procyanidin B2	5.47	275	577	0.00	71.49 ± 9.0 ^a^	0.09 ± 0.00 ^b^	0.00	0.03 ± 0.00 ^b^	0.01 ± 0.00 ^b^	0.01 ± 0.00 ^b^
(−)-epicatechin	5.90	280	289	0.00	56.7 7± 4.1 ^a^	0.07 ± 0.00 ^b^	0.00	0.04 ± 0.00 ^b^	0.01 ± 0.00 ^b^	0.00
Procyanidin C1	5.98	280	866	0.00	22.01 ± 2.6 ^a^	0.00	0.00	0.00	0.00	0.00
Sum	-	-	-	0.00	170.84 ± 12.4 ^a^	0.16 ± 0.00 ^b^	0.0	0.07 ± 0.00 ^b^	0.02 ± 0.00 ^b^	0.01 ± 0.00 ^b^
Total of polyphenols	-	-	-	1542.89 ± 135 ^b^	235.98± 19.6 ^d^	2336.55 ± 212 ^a^	722.58 ± 72.4 ^c^	2528.61 ± 227 ^a^	2353.55 ± 231 ^a^	2257.46 ± 199 ^a^

^1^ Identification confirmed by commercial standards; ^2^ experiment data; mean values with different letters (a–e) within same row were statistically different (*p* = 0.05). Values expressed as mean ± standard deviation.

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
