# Peer review of "Effect of Vacuum Impregnation with Apple-Pear Juice on Content of Bioactive Compounds and Antioxidant Activity of Dried Chokeberry Fruit"

_foods, 2020, doi:10.3390/foods9010108_

Round 1

Reviewer 1 Report

Dear authors,

after the revision of the manuscript “Effect of vacuum impregnation with apple-pear juice on content of bioactive compounds and antioxidant activity of dried chokeberry fruit” several modifications in my opion must be done before pubblication.

- Introduction

Since the apple-pear juice impregnation represent the most important aspect of the work, as also highlighted in the title, is necessary to add a paragraph on the bioactives content in fruit juices (please see manuscript on the fruit juices characterization as:

1) Rojas-Garbanzo, C., Winter, J., Montero, M. L., Zimmermann, B. F., & Schieber, A. (2019). Characterization of phytochemicals in Costa Rican guava (Psidium friedrichsthalianum-Nied.) fruit and stability of main compounds during juice processing-(U) HPLC-DAD-ESI-TQD-MSn. Journal of Food Composition and Analysis, 75, 26-42;

2) Difonzo, G., Vollmer, K., Caponio, F., Pasqualone, A., Carle, R., & Steingass, C. B. (2019). Characterisation and classification of pineapple (Ananas comosus [L.] Merr.) juice from pulp and peel. Food control, 96, 260-270).

- Methods

It is necessary to add specifications of juices used for impregnation, and also a characterization of main bioactive compounds, since as you conclude in the manuscript the final antioxidant activity and phenolics profile are the result of the juice impregnation.

Moreover, different modifications are requested on the whole manuscript, that I’ve reported directly in the pdf text, thus please consider the attached pdf with the comments.

An English revision is requested.

Author Response

Since the apple-pear juice impregnation represent the most important aspect of the work, as also highlighted in the title, is necessary to add a paragraph on the bioactives content in fruit juices (please see manuscript on the fruit juices characterization as:

1) Rojas-Garbanzo, C., Winter, J., Montero, M. L., Zimmermann, B. F., & Schieber, A. (2019). Characterization of phytochemicals in Costa Rican guava (Psidium friedrichsthalianum-Nied.) fruit and stability of main compounds during juice processing-(U) HPLC-DAD-ESI-TQD-MSn. Journal of Food Composition and Analysis, 75, 26-42;

The properties of apple-pear juice were added in the introduction, the above publication was used.

It is necessary to add specifications of juices used for impregnation, and also a characterization of main bioactive compounds, since as you conclude in the manuscript the final antioxidant activity and phenolics profile are the result of the juice impregnation.

The specification of the juice used is given in the tables 1 and 2.

Not all notes can be read: l. 117; 138; 152; 180; 300

l. 255 and 256 were removed
An "equivalent" in unit has been added

Header change in the table

Other minor comments have also been corrected

Reviewer 2 Report

Overall, this paper is straightforward in its methods and results. However, the purpose and value of the study remains unclear. This must be addressed.

Line 53

Be consistent with terminology….use either “procyanidins” or “proanthocyanidins”, but only use one of these terms throughout

Line 59

The term “confiture” should be replaces with “preserves”

Line 86-89

It is not clear what the purpose of the study is…a lot of the introduction discusses drying, but this is not mentioned in the aim of the study…is the aim to offset losses during drying by adding apple pear juice?

Why were chokeberries selected? Are they particularly susceptible to polyphenol losses during drying?

Why was apple-pear juice selected?

A lot of these questions need to be clarified.

Line 109-112

Were the fruits in season?

How long were fruits stored prior to use?

Line 116-118, Line 244-245, 262-263, 283,

The characteristics of the pear-apple juice need to be described. The goals of the project remain somewhat unclear, which hinders my ability to evaluate the data.

Line 119

Was there a control group with 0 kPa pressure? This would be an important control

Line 198, 292

I think this should say “within the same row”, not ““within the same column”

Line 326-327

“Retention” is not the proper word…”addition” or other words would be appropriate

Author Response

Overall, this paper is straightforward in its methods and results. However, the purpose and value of the study remains unclear. This must be addressed.

The purpose and value of the study have been clarified.

line 53

Be consistent with terminology….use either “procyanidins” or “proanthocyanidins”, but only use one of these terms throughout

has been corrected

Line 59

The term “confiture” should be replaces with “preserves”

after language correction MPDI remained "confitures"

Line 86-89

It is not clear what the purpose of the study is…a lot of the introduction discusses drying, but this is not mentioned in the aim of the study…is the aim to offset losses during drying by adding apple pear juice?

the purpose of the study was clarified, and the main purpose was to improve the taste of dried chokeberry fruit

Why were chokeberries selected? Are they particularly susceptible to polyphenol losses during drying?

Aronia is rich in anthocyanins but is astringent to reduce the astringent taste, apple-pear juice has been added. Polyphenols are sensitive to high temperatures in the long time that is drying. This was added in the text.

Why was apple-pear juice selected?

Apple-pear juice was used for the impregnation due to improve the tart taste of chokeberry fruits by introducing sugars, acids, aroma and other sensoric components of apples and pears.

A lot of these questions need to be clarified.

Line 109-112

Were the fruits in season?

Yes, in 2018, this has been explained

How long were fruits stored prior to use?

The juice was pressed immediately after harvesting and kept refrigerated until tested. The chokeberry fruit after harvesting was stored for 1 week under refrigerated conditions

Line 116-118, Line 244-245, 262-263, 283,

The characteristics of the pear-apple juice need to be described. The goals of the project remain somewhat unclear, which hinders my ability to evaluate the data.

The purpose of the work has been improved, the characteristics of apple-pear juice in the tables 1 and 2 have been added

Line 119

Was there a control group with 0 kPa pressure? This would be an important control

Unfortunately, no such tests were performed. The authors concluded that testing at 3 pressures and row materials are sufficient

Line 198, 292

I think this should say “within the same row”, not ““within the same column”

has been corrected

Line 326-327

“Retention” is not the proper word…”addition” or other words would be appropriate

has been corrected

Round 2

Reviewer 1 Report

The revision is OK